# The Zero-Velocity Correction Method for Pipe Jacking Automatic Guidance System Based on Fiber Optic Gyroscope

**DOI:** 10.3390/s24185911

**Published:** 2024-09-12

**Authors:** Wenbo Zhang, Lu Wang, Yutong Zu

**Affiliations:** School of Engineering and Technology, China University of Geosciences, Beijing 100083, China; 2102220066@email.cugb.edu.cn (W.Z.); 3002220025@email.cugb.edu.cn (Y.Z.)

**Keywords:** TCZVD, FOG-INS, pipe jacking, Kalman filter

## Abstract

The pipe jacking guidance system based on a fiber optic gyroscope (FOG) has gained extensive attention due to its high degree of safety and autonomy. However, all inertial guidance systems have accumulative errors over time. The zero-velocity update (ZUPT) algorithm is an effective error compensation method, but accurately distinguishing between moving and stationary states in slow pipe jacking operations is a major challenge. To address this challenge, a “MV + ARE + SHOE” three-conditional zero-velocity detection (TCZVD) algorithm for the fiber optic gyroscope inertial navigation system (FOG-INS) is designed. Firstly, a Kalman filter model based on ZUPT is established. Secondly, the TCZVD algorithm, which combines the moving variance of acceleration (MV), angular rate energy (ARE), and stance hypothesis optimal estimation (SHOE), is proposed. Finally, experiments are conducted, and the results indicate that the proposed algorithm achieves a zero-velocity detection accuracy of 99.18% and can reduce positioning error to less than 2% of the total distance. Furthermore, the applicability of the proposed algorithm in the practical working environment is confirmed through on-site experiments. The results demonstrate that this method can effectively suppress the accumulated error of the inertial guidance system and improve the positioning accuracy of pipe jacking. It provides a robust and reliable solution for practical engineering challenges. Therefore, this study will contribute to the development of pipe jacking automatic guidance technology.

## 1. Introduction

The pipe jacking method, a common trenchless pipeline installation technique, is widely used in urban pipeline construction due to its minimal excavation and low environmental impact [1]. With the increasing complexity of underground utilities and the growing demand for long-distance pipe jacking, precise measurement of the position and orientation of the pipe jacking machine is required to ensure that the machine head jacks along the predetermined trajectory, thereby avoiding complex underground structures [2,3].

Currently, common pipe jacking guidance methods include the laser target guidance system and the prism guidance system. The laser target guidance system suffers from accuracy issues due to the divergence of the laser beam over long distances, while the prism guidance system often faces difficulties in prism locking. Therefore, traditional measurement schemes are not suitable for long-distance pipe jacking. The gyroscope-based inertial navigation method can address long-distance pipe jacking guidance issues. This method primarily relies on the outputs from a three-axis accelerometer and a three-axis gyroscope, combined with inertial navigation algorithms, to calculate the position, velocity, and attitude of the carrier, offering a high degree of autonomy [4,5]. As a result, an increasing number of enterprises are applying inertial navigation technology to pipe jacking. Notable examples include the TUnIS Navigation MT gyroscope system developed by Germany’s VMT GmbH, the TMG-32B navigation system developed by Japan’s Keiki Corporation, and the curved pipe jacking navigation system developed by China’s Shanghai Lixin Company. However, all inertial navigation systems suffer from the issue of accumulating errors over time, which can lead to significant positioning inaccuracies over prolonged periods [6,7].

Among the various error correction methods, the zero-velocity update (ZUPT) algorithm has proven to be an effective solution. It uses the zero-velocity information of the carrier during stationary periods to periodically reset the residual velocity to zero, thereby suppressing velocity output errors [8,9,10]. This method has been widely applied in indoor pedestrian navigation. For example, Li et al. [11] achieved more accurate position estimation by using an improved ZUPT combined with an extended Kalman filter (EKF). Similarly, B. Wagstaff et al. [12] improved the accuracy of a foot-mounted zero-velocity aided inertial navigation system (INS) by adjusting the zero-velocity detection threshold through a real-time motion classifier. For pipe jacking, during the addition of each new pipe segment, the pipe jacking machine remains stationary. This periodic static process provides a basis for zero-velocity correction.

Before implementing the ZUPT algorithm, it is essential to perform zero-velocity detection on the pipe jacking machine to determine when the system is stationary. Thus, the stationary state is a prerequisite for the ZUPT algorithm, and its performance highly depends on the accuracy of zero-velocity detection. Extensive research has been conducted in this area, with mainstream methods primarily including threshold-based detection and deep learning-based detection. Threshold-based detection methods can be further divided into fixed threshold and adaptive threshold methods.

Fixed threshold methods typically involve predefined thresholds and sliding windows. The state statistics are calculated based on the output from the inertial measurement unit (IMU), and when these statistics fall below the threshold, a zero-velocity condition is assumed, transforming zero-velocity detection into a binary hypothesis detection problem [13]. Typical methods include: (1) an acceleration-moving variance (MV) detector [14]; (2) an acceleration magnitude (MAG) detector [15,16]; (3) an angular rate energy (ARE) detector [17,18,19]; and (4) a stance hypothesis optimal estimation (SHOE) detector [20] (combining acceleration amplitude and angular velocity energy). Among these, the SHOE detector has proven to be particularly effective and is widely used in fixed threshold detection methods [21]. Additionally, a multi-conditional approach can be employed for zero-velocity detection, where zero velocity is determined only if all designed conditions are met, further improving the accuracy of zero-velocity detection [22,23,24].

Adaptive threshold methods primarily focus on establishing adaptive threshold adjustment models. Tian [25] and Wang [26] developed adaptive threshold adjustment models based on gait frequency and gait velocity, respectively. Skog et al. [27] designed an adaptive version of SHOE using Bayesian adaptive estimation. Sun et al. [28] proposed an adaptive ZUPT technique based on specific energy consumption (SEC) curves for various motion patterns. Li et al. [29] utilized statistical features of detection measures (such as peaks, variance, and averages) to dynamically adjust thresholds. These methods have shown impressive performance in the literature. However, their algorithm designs are often more complex and are mainly used in scenarios with varied motion patterns, such as running, walking, and stair climbing in pedestrian navigation. For pipe jacking, the practical working environment is characterized by low-speed conditions and relatively simple motion patterns, either jacking or stopping. Thus, it typically does not involve significant or high-frequency threshold variations.

With the development of artificial intelligence, threshold-free detection methods based on deep learning offer new perspectives for zero-velocity detection. Chen [30] designed a convolutional neural network zero-velocity detector (CNN-ZVD) based on deep learning and demonstrated its significant advantages in composite motion scenarios. Yang [31] proposed a deep recurrent convolutional neural network (RCNN) with a symmetric network for adaptive zero-velocity interval detection, further improving model performance. Ran Wei [32] developed an adaptive zero-velocity update (AZUPT) method based on convolutional neural networks (CNNs) to classify ZUPT conditions. Although these methods can achieve better performance, they require extensive data for training, resulting in significant workload, long processing times, and high costs. Additionally, deep learning models often suffer from overfitting [33]. Given the current difficulty in obtaining large amounts of data on the practical motion of pipe jacking machines, fixed threshold zero-velocity detectors are considered more suitable for this application.

Therefore, a “MV + ARE + SHOE” three-condition zero-velocity detection (TCZVD) algorithm for a fiber optic gyroscope-inertial navigation system (FOG-INS) is developed in this study. This algorithm integrates acceleration-moving variance, angular rate energy, and stance hypothesis optimal estimation detectors for accurate zero-velocity detection. Firstly, a pipe jacking positioning error correction model is established. Through simulated experiments, the algorithm achieves a high zero-velocity detection accuracy of 99.18%. Based on these results, velocity errors are corrected, ultimately reducing positioning errors to within 2% of the total distance. Its positioning accuracy is better than any of the position estimation results when the other three zero-velocity detectors are used individually. Additionally, practical pipe jacking experiments confirm the algorithm’s capability to sensitively detect pipe jacking moving and stationary states, thereby promoting the application of zero-velocity correction technology in pipe jacking and further improving the guidance and positioning accuracy of pipe jacking.

## 2. Moving Characteristic Analysis for Pipe Jacking

The pipe jacking construction method is shown in Figure 1, in which the jacking cylinder is installed in the working well, and the pipe jacking machine is positioned in front of the cylinder. The machine excavates the soil while hydraulic cylinders provide thrust, pushing the machine through the soil from the working well to the receiving well. Simultaneously, pipe segments are sequentially laid between the two wells. 

Pipe jacking takes about 2~3 h for each segment of pipe (3 m) and usually stops for the installation of the next pipe segment. The process of installing a pipe segment typically lasts 10 to 15 min. During this shutdown period, construction personnel need to disconnect the cables before installing the pipe segment. Simultaneously, the pipe jacking machine temporarily halts its thrust and rotational movements, creating a suitable environment for zero-velocity correction. During this time, using the measurement data of FOG-INS at the head of the pipe jacking machine, the stationary moment of the head can be monitored in real time. The zero velocity-correction algorithm can be executed at this moment to calibrate sensor biases, thereby improving the measurement accuracy of the navigation system. 

After installation of the new pipe segment, construction personnel restart the hydraulic cylinder, power up the pipe jacking machine, and resume pushing forward. Due to the completion of zero-velocity correction during the downtime, the machine can directly return to its jacking operation. This cyclic motion continues until the pipe jacking machine reaches the reception well, completing the entire pipeline installation.

Overall, the periodic stopping of the pipe jacking machine for pipe segment updates provides the ideal conditions for zero-velocity correction. Utilizing the downtime for zero-velocity correction not only enhances the accuracy of the navigation system but also improves project efficiency.

## 3. The Zero-Velocity Correction Model

### 3.1. The Kalman Filter Model Based on Zero-Velocity Correction

During the inertial navigation of pipe jacking, inherent measurement errors of the sensors affect navigation accuracy, so an error model is needed to correct the navigation states.

The discrete system state-space equations and measurement equations are represented as follows:(1)$$\left\{\begin{array}{l} \overset{•}{X_{k}}=F_{k}X_{k-1}+\omega_{k-1}\\ Z_{k}=HX_{k}+\upsilon_{k} \end{array}\right.$$

The error state vector used in this study consists of 15 dimensions, defined as follows:(2)$$X={\left[\begin{array}{ccccc} \delta p^{n} & \delta v^{n} & \delta a^{n} & \delta g^{b} & \delta w^{b} \end{array}\right]}^{T}$$
where $\delta p^{n},\delta v^{n},\delta a^{n}$ are the three-dimensional position, velocity, and attitude error vectors in the navigation coordinate system, respectively. The bias error vectors of the accelerometer and gyroscope are represented by $\delta g^{b}$ and $\delta w^{b}$, respectively.

In the state-space equation, $X_{k}$ represents the error state vector of the system at time *k*, $\omega_{k-1}$ denotes the process noise of the system, and $F_{k}$ is the state transition matrix of the system. Its specific definition can be expressed as follows:(3)$$F_{k}=\left[\begin{array}{ccccc} I_{3\times 3} & \Delta t\cdot I_{3\times 3} & 0_{3\times 3} & 0_{3\times 3} & 0_{3\times 3}\\ 0_{3\times 3} & I_{3\times 3} & \Delta t\cdot S_{k} & \Delta t\cdot C_{b}^{n} & 0_{3\times 3}\\ 0_{3\times 3} & 0_{3\times 3} & I_{3\times 3} & 0_{3\times 3} & -\Delta t\cdot C_{b}^{n}\\ 0_{3\times 3} & 0_{3\times 3} & 0_{3\times 3} & I_{3\times 3} & 0_{3\times 3}\\ 0_{3\times 3} & 0_{3\times 3} & 0_{3\times 3} & 0_{3\times 3} & I_{3\times 3} \end{array}\right]$$

The terms $I_{3\times 3}$ and $0_{3\times 3}$ represent three-dimensional diagonal matrices with ones and zeros on the diagonal, respectively. Δt represents the discretized time interval of the system, and $C_{b}^{n}$ is the rotation matrix from the b-frame to the n-frame. The skew-symmetric matrix ($S_{k}$) is composed of the specific force information measured by the accelerometer in the navigation coordinate system and can be defined as follows:(4)$$S_{k}=\left[\begin{array}{ccc} 0 & -f_{k}^{n}(3) & f_{k}^{n}(2)\\ f_{k}^{n}(3) & 0 & -f_{k}^{n}(1)\\ -f_{k}^{n}(2) & f_{k}^{n}(1) & 0 \end{array}\right]$$
When the system detects a zero-velocity state, this information can be used as a measurement to perform a zero-velocity update on the strapdown inertial navigation system, further correcting the system’s navigation state outputs. At this time, the measurement error $Z_{k}$ at time *k* can be calculated by the following formula: (5)$$Z_{k}=v_{k}^{-}-\left[\begin{array}{ccc} 0 & 0 & 0 \end{array}\right]$$
where $v_{k}^{-}$ is the prior estimate of velocity at time *k*. In Formula (1), $\upsilon_{k}$ represents the measurement noise of the system, and ***H*** is the measurement matrix, defined in the ZUPT model as follows:(6)$$H=\left[\begin{array}{ccccc} 0_{3\times 3} & I_{3\times 3} & 0_{3\times 3} & 0_{3\times 3} & 0_{3\times 3} \end{array}\right]$$

In the experiment, the Kalman filter parameters can be adjusted according to the practical situation to optimally estimate the system-state error, and then the estimated error is fed back to the strapdown inertial navigation algorithm module, so as to achieve further correction of the system navigation parameters and inertial data.

Considering the instability of the INS vertical channel, external sensors such as a level instrument are typically required in practical pipe jacking applications to perform altitude measurements, thereby suppressing the divergence in inertial navigation altitude calculations. Assuming that the vertical channel information from the level instrument is known, the measurement matrix H is modified from the original 3 × 15 matrix to a 4 × 15 matrix, as shown below:(7)$$H=\left[\begin{array}{ccccc} \left[\begin{array}{c} \begin{array}{ccc} 0 & 0 & 1 \end{array}\\ 0_{3\times 3} \end{array}\right] & \left[\begin{array}{c} \begin{array}{ccc} 0 & 0 & 0 \end{array}\\ I_{3\times 3} \end{array}\right] & 0_{4\times 3} & 0_{4\times 3} & 0_{4\times 3} \end{array}\right]$$

The error state estimation X of the system can be calculated by the following formula: (8)$$X_{k}=K\cdot \left[\begin{array}{c} p_{k}^{-}(3)\\ v_{k}^{-} \end{array}\right]$$
where **K** represents the Kalman gain, and $p_{k}^{-}(3)$ is the height information at time *k*. Thus, more accurate height information can be obtained.

### 3.2. Zero-Velocity Detection Algorithm Model

The prerequisite for zero-velocity correction is that the system is in a zero-velocity state, so it is necessary to accurately detect the zero-velocity moment before the ZUPT algorithm can be executed. According to the above analysis of the moving characteristics of pipe jacking, a fixed threshold zero-velocity detector is considered more suitable for the application scenario of pipe jacking. Hence, a three-condition combined zero-velocity detection method for FOG-INS was designed for the pipe jacking process.

The inertial sensor infers the motion state of the object according to the principle of inertia, so the motion state can be deduced by the data output of the inertial device. When the carrier is in a static state, the theoretical value of the acceleration vector should be close to the gravity acceleration, that is $g=9.81\;m/s^{2}$, and the angular rate should be close to zero. Additionally, the data fluctuation of the sensor should be less than the moving state in the static state. Therefore, the sliding window *W* and respective threshold *γ* are defined for different zero-velocity detectors, and the detection statistic *T* is calculated. When it is less than the threshold, the system is considered to be in a zero-velocity state. To improve detection accuracy, a three-condition combined zero-velocity detection method is adopted, defined as follows:

Condition 1 (C1): Acceleration-Moving Variance (MV) Detector.
(9)$$C1=\left\{\begin{array}{l} 1,T=\frac{1}{\sigma_{a}^{2}W}\sum_{k=n}^{n+W-1}||y_{k}^{a}-\bar{y_{n}^{a}}|{|}^{2}<\gamma_{a}\\ 0,Others \end{array}\right.$$

Among them, $y_{k}^{a}$ is the acceleration at time *k*, $y_{k}^{a}=\sqrt{f_{k}{\left(x\right)}^{2}+f_{k}{\left(y\right)}^{2}+f_{k}{\left(z\right)}^{2}}$ where $f_{k}(x)$, $f_{k}(y)$, $f_{k}(y)$ represent the three-axis acceleration-specific force information measured by the accelerometer at time *k*, respectively. $\bar{y_{n}^{a}}$ represents the sample mean of the accelerometer in the sliding window, and $\gamma_{a}$ is the threshold for the moving variance detector. When *T* is less than the threshold, the pipe jacking is considered stationary.

Condition 2 (C2): Stance Hypothesis Optimal Estimation (SHOE) Detector.
(10)$$C2=\left\{\begin{array}{l} 1,T=\frac{1}{W}\sum_{k=n}^{n+W-1}\left(\frac{1}{\sigma_{a}^{2}}|\right|y_{k}^{a}-g\frac{\bar{y_{n}^{a}}}{\left||\bar{y_{n}^{a}}|\right|}|{|}^{2}+\frac{1}{\sigma_{\omega }^{2}}||y_{k}^{\omega }|{|}^{2})<\gamma \\ 0,Others \end{array}\right.$$

In the formula, both the acceleration and angular velocity information are combined, and $y_{k}^{\omega }$ represents the angular rate at time *k*, $y_{k}^{w}=\sqrt{\omega_{k}{\left(x\right)}^{2}+\omega_{k}{\left(y\right)}^{2}+\omega_{k}{\left(z\right)}^{2}}$, where $\omega_{k}(x)$, $\omega_{k}(y)$$\omega_{k}(z)$ are the three-axis angular rates measured by the fiber optic gyroscope at time *k*, respectively. This condition assumes that the acceleration magnitude and angular rate energy should be sufficiently small to consider the pipe jacking to be in a stationary state.

Condition 3 (C3): Angular Rate Energy (ARE) Detector.
(11)$$C3=\left\{\begin{array}{l} 1,T=\frac{1}{\sigma_{\omega }^{2}W}\sum_{k=n}^{n+W-1}\left|\right|y_{k}^{\omega }|{|}^{2}<\gamma_{\omega }\\ 0,Others \end{array}\right.$$

$\gamma_{\omega }$ represents the threshold of the angular velocity energy detector, which considers that when the angular velocity energy is less than the threshold, the pipe jacking is in a static state.

It is worth noting that in the above three equations, the sliding window size *W* is set to 100, and $\sigma_{a}$ and $\sigma_{\omega }$ represent the noise variance of the accelerometer and gyroscope, respectively, which only scale the test statistic in conditions C1 and C3. However, the noise ratio of $\sigma_{a}^{2}/\sigma_{\omega }^{2}$ will affect the performance of the zero-velocity detector in condition C2, which represents the data disturbance from the accelerometer and gyroscope. 

The zero-velocity moment can be judged only if the above three zero-velocity conditions are simultaneously satisfied, that is: (12)$$C_{i}=C1\cap C2\cap C3$$

The zero-velocity detection process of the system is shown in Figure 2 below:

Among them, L represents the window size of median filtering, which defines the range of single filtering and retains certain edge information while ensuring the smoothness of filtering, in this study, L = 100. Firstly, the three zero-velocity detectors are set to appropriate thresholds and sliding windows. Then, using the acceleration and angular velocity data collected by the FOG-INS, the state statistic T of the three zero-velocity detectors within the window range are calculated, respectively. These statistics are evaluated to determine if they are simultaneously below the set thresholds, indicating that all three conditions are met. Once these conditions are satisfied, median filtering is applied to filter out the sensor noise to obtain the final zero-velocity detection outcome.

## 4. Experiment and Analysis

To evaluate the accuracy of the proposed TCZVD algorithm and to verify its feasibility in detecting the moving/stationary state of the pipe jacking machine in real working conditions, two sets of experiments were designed. The first set of experiments used a simulation platform to regularly simulate the state changes of jacking and stopping of linear pipe jacking, in order to verify the performance of the proposed zero-velocity detection algorithm and the accuracy of the positioning model. The second set of experiments focused on the low-speed environment of practical pipe jacking operations to validate the feasibility and robustness of the algorithm in monitoring the motion state of the pipe jacking machine.

### 4.1. Simulation Experiment

#### 4.1.1. Experimental Platform Setup

The fiber optic gyroscope inertial navigation system (FOG-INS) used in the experiment includes a three-axis accelerometer and a three-axis fiber optic gyroscope. The performance parameters of the accelerometer and gyroscope are shown in Table 1. This FOG-INS system utilizes RS-422 serial communication with a sampling frequency of 100 Hz and a baud rate of 115,200 bits/s. Additionally, a wheel odometer is used, which employs RS-485 for data transmission with a sampling frequency of 1 Hz.

Before the experiment, a linear trajectory was pre-designed to simulate the linear jacking trajectory of the pipe jacking machine. The FOG-INS system is fixed at the center of a trolley, ensuring that the *Y*-axis of the FOG-INS is parallel to the designed trajectory. The odometer is mounted on the wheel of the trolley so that it can be tightly attached to the ground to prevent slipping. The specific setup is shown in Figure 3. During the simulation of pipe jacking, the trolley is manually pushed from the starting point, resting for 30 s after every 30 s of jacking, and then continuing this cycle until reaching the end of the designed trajectory. It is important to note that, due to the manual pushing method used in the simulation, each movement and rest interval cannot be strictly maintained at 30 s. Therefore, the odometer’s calculated results are used as reference values to verify the accuracy and reliability of the proposed algorithm in zero-velocity detection and position estimation.

#### 4.1.2. Experimental Results Analysis

Firstly, the odometer data obtained from the experiment are further processed to obtain the estimated position results, as shown in Figure 4. The figure shows that the designed trajectory length is 36.55 m, with a total duration of 1921s. The relationship graph between distance and time exhibits a step-like pattern, which visually represents the moving and stationary states of the simulated pipe jacking process. In this graph, the horizontal line segments represent the stationary phases, while the oblique line segments indicate the jacking phases.

To accurately detect the zero-velocity moments and their duration in the simulated pipe jacking process, the relationship between speed and time is derived based on odometer data. As shown in Figure 5, red dots represent detected zero-velocity points. When zero velocity is confirmed, the zero-velocity detector is set to 1, otherwise, it is set to 0, generating the bar-shaped zero-velocity detection result graph depicted. As can be seen from the figure, the jacking moving and stationary states presented by the odometer data match the simulated jacking motion pattern of the intended design.

Considering this is a binary classification problem, in order to facilitate the quantitative analysis of the proposed zero-velocity detector’s performance, this study adopted analysis indexes commonly used in the two-classification: Precision, Recall, and $F_{\beta }$, which takes into account both precision and recall. The specific calculation formulas are as follows:(13)$$Precision=\frac{TP}{TP+FP}$$
(14)$$Recall=\frac{TP}{TP+FN}$$
(15)$$F_{\beta }=(1+\beta^{2})\times \frac{Precision\times Recall}{\beta^{2}\times Precision+Recall}$$
where *TP* represents the number of true positives, *FP* represents the number of false positives, *FN* represents the number of false negatives, and *TN* represents the number of true negatives. The parameter *β* represents the weight. The parameter $F_{\beta }$, ranging from 0 to 1, considers both Precision and Recall, with higher values indicating better model performance.

Thus, precision indicates the proportion of true positive cases among those predicted as positive by the model. A high precision means fewer false positives. Recall represents the proportion of practical positive cases correctly predicted by the model, with a high recall indicating fewer false negatives. Considering that in the context of zero-velocity detection, false positives significantly impact navigation performance because they imply that non-zero-velocity points are incorrectly identified as zero-velocity points. This incorrect correction in the inertial navigation solution can lead to greater errors. Therefore, in this experiment, more emphasis was placed on improving precision. It is known that when *β* < 1, precision has a greater influence; when *β* > 1, recall has a greater influence; and when *β* = 1, it represents the standard F1 score, which is the harmonic mean of precision and recall. To highlight the importance of precision, the F0.5 score is used to evaluate the detection model’s performance.

During the simulated jacking process, the FOG-INS continuously collects data. The raw data from the FOG-INS, shown in Figure 6, indicate that both the three-axis fiber optic gyroscope and the three-axis accelerometer are sensitive to the moving state of the simulated pipe jacking. The data fluctuations during movement are significantly higher than during stationary periods, and this feedback aligns well with the practical pipe jacking moving state.

Figure 7 shows the overall detection results of the proposed three-condition combined zero-velocity detection (TCZVD) algorithm integrated with median filtering. It is worth mentioning that the test statistics in the figure refer to the statistics calculated from the output data of the FOG-INS, which is used to determine whether the carrier is in a stationary state. It can reflect the moving characteristics of the carrier within a specific time window.

As shown in Figure 7, the zero-velocity detection results of this algorithm closely match the odometer results in Figure 5. It effectively and accurately identifies the moving state of the simulated pipe jacking. 

In this experiment, the odometer’s zero-velocity detection results are used as a reference. A quantitative comparison is conducted using the first 10 min of experimental data, as illustrated in Figure 8.

With further quantitative analysis from Figure 8, the confusion matrix can be obtained, as shown in Figure 9.

From Figure 9, it is evident that the designed TCZVD algorithm achieves a precision rate of 99.18%. The recall rate is relatively lower, resulting in an F0.5 score of 95.13%. However, as previously analyzed, the zero-velocity detection pays more attention to the index of precision, and missed detections have a relatively minor impact on the guidance system. Therefore, the detector is considered to perform well in zero-velocity detection.

When the system detects a zero-velocity state, the next step is to perform the ZUPT algorithm. It is known that the simulated jacking experiment involves jacking for 30 s and resting for 30 s. In practical operations, the jacking process takes about 15 min for pipe segment installation. Considering the possible false detections in the zero-velocity detector, and in order to ensure that the detected zero-velocity points are true zero-velocity points so as to improve the accuracy of the jacking position estimation, it is considered that only continuous 3 s detection at zero velocity can be considered as a zero-velocity period. Figure 10 shows the estimated positions of the simulated jacking process based on the detection results of the various zero-velocity detectors.

For distance estimation, the reference distance $X_{ref}$ provided by the odometer is 36.55 m. The formula for calculating the relative position error rate is as follows:(16)$$\eta =\frac{\left|X_{ref}-X\right|}{X_{ref}}$$

Using the three-condition combined zero-velocity detection algorithm, the final position estimation results are shown in Figure 11. 

As shown in the figure above, the distance and time curve estimated by the algorithm presented in this study closely approximates the reference true value. The specific distance estimates for each zero-velocity detector are shown in Table 2. From the table, it is evident that the three-condition combined zero-velocity detection algorithm used in this study has improved the accuracy to 98.06%, with a relative error rate of 1.94%. It demonstrates a higher position estimation accuracy compared to other detectors.

In the pipe jacking simulation experiment, the positioning accuracy of the pipe jacking has a certain relationship with the regularity of the moving time, which means that the ratio of moving and stationary time will affect the positioning accuracy to a certain extent. This is because the positioning error of INS will accumulate over time, and ZUPT cannot correct the accumulation error in the non-zero speed stage. Therefore, it was necessary to design experiments to observe how the positioning accuracy of the TCZVD algorithm changes with the different proportions of moving and stationary time.

In the experiment, the position estimation result of the odometer is still used as the reference value. It is known that the final jacking distance of the odometer is about 35.56 m. The position estimation results of each zero-speed detector are observed by continuously changing the proportion of moving and stationary time of the simulated pipe jacking. The estimation of the whole position of the pipe jacking according to the results of the above zero-speed detectors are shown in Figure 12.

The specific positional estimates for each zero-speed detector are shown in Table 3.

As can be seen from the table, under the condition of constantly changing the proportion of moving and stationary time of the simulated pipe jacking, when the three zero-velocity detectors of MV, ARE, and GLRT are detected independently, the position estimation results obtained by the two algorithms of MV and ARE ultimately are slightly reduced, while the position estimation accuracy of GLRT is improved. Although the position estimation accuracy calculated by the TCZVD algorithm designed in this study is slightly reduced, it still maintains 97.61% position estimation accuracy, which further proves that the TCZVD algorithm designed in this paper can effectively avoid the limitations of a single detector. Finally, the reliability and robustness of detection are improved.

### 4.2. Field Experiment

#### 4.2.1. Experimental Platform

To further verify that the designed algorithm can accurately distinguish between the moving and stationary states of the pipe jacking machine in low-speed scenarios, on-site experiments were conducted. The cutter head of the pipe jacking machine is shown in Figure 13, and the FOG-INS is installed inside the pipe jacking machine, as shown in Figure 14.

After the sensors are installed, the fiber optic inertial navigation system performs the initial alignment before the pipe jacking machine is turned on. Then, the pipe jacking machine is started, so that the cutter rotated for a period of time before shutting down. During this process, the FOG-INS continuously collects data.

#### 4.2.2. Experimental Processing Results

The data collected by the FOG-INS are further processed. The raw data from the three-axis accelerometer and three-axis gyroscope are shown in Figure 15.

The raw measurement data from the FOG-INS, shown in Figure 14, clearly indicate significant changes in all three directions when the pipe jacking machine is started. This verifies that even in slow pipe jacking operations, the three-axis accelerometer and three-axis gyroscope are sensitive to the pipe jacking machine’s movements. Using the data of FOG-INS to predict the pipe jacking machine’s moving state is a feasible technical approach. From the raw data, it is noted that the initial alignment of the pipe jacking machine takes approximately 13.7 min, the cutter rotation lasts around 11.4 min before stopping, and the entire process lasts about 28.2 min. The zero-velocity detection results, obtained using the proposed zero-velocity detection algorithm, are shown in Figure 16.

A comparison of Figure 15 and Figure 16 demonstrates that the proposed algorithm can accurately detect the moments of moving and stationary states of the pipe jacking machine in practical working conditions. This further validates the feasibility of the designed zero-velocity detection algorithm for application in pipe jacking operations. 

## 5. Conclusions and Prospects

To address the issue of error accumulation over time in pipe jacking inertial guidance systems, this study aims to apply zero-velocity update (ZUPT) algorithm to pipe jacking to further improve its guidance and positioning accuracy. Before performing ZUPT, it is crucial to determine whether the system is in a zero-velocity state, so zero-velocity detection is particularly important. Based on this, the study introduces a “MV + ARE + SHOE” three-conditional zero-velocity detection (TCZVD) algorithm. This algorithm employs the acceleration and angular velocity measurements from the FOG-INS and combines three conditions to identify zero-velocity states of the pipe jacking machine.

Experimental results demonstrate that the designed three-condition zero-velocity detection algorithm achieves a zero-velocity detection accuracy of 99.18%, with positioning errors below 2% of the total distance. And even under the condition of constantly changing the proportion of simulated pipe jacking moving and stationary times, the position estimation results obtained according to the TCZVD algorithm perform the best compared to the other three detectors when they are detected independently, and the positioning error is kept within 3% of the total distance. Furthermore, on-site experiments of pipe jacking confirmed that the designed zero-velocity detection algorithm can sensitively detect the motion changes of the pipe jacking machine during practical operations. This not only provides a reliable technical solution to the problem of error accumulation in pipe jacking inertial guidance systems but also improves the accuracy and reliability of pipe jacking guidance technology. Consequently, this development advances the technology towards greater intelligence and efficiency, laying the foundation for automated pipe jacking measurements.

Regarding the application technology of zero-velocity correction in pipe jacking, there are still many directions to be explored in the future. Firstly, the combination of machine learning algorithms and zero-velocity detection technology can be considered to further improve the intelligence of the system. Secondly, the adaptability and stability of the technology under different working conditions can be explored to extend its application range. Finally, how to integrate the technology with other sensors can be studied to achieve a more accurate and comprehensive pipe jacking guidance system. Through these studies, pipe jacking guidance technology is expected to be more widely used in future engineering practices and promoted in the development of related fields.

## Figures and Tables

**Figure 1 sensors-24-05911-f001:**
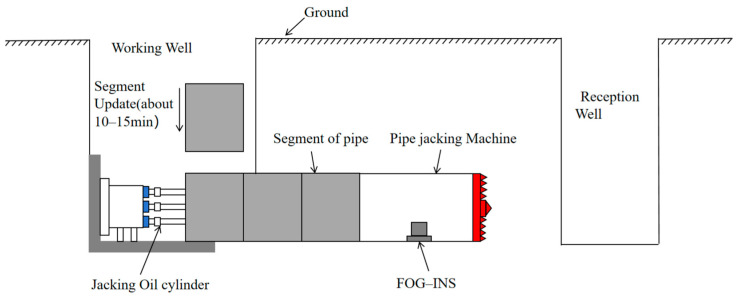
The construction technique of pipe jacking.

**Figure 2 sensors-24-05911-f002:**
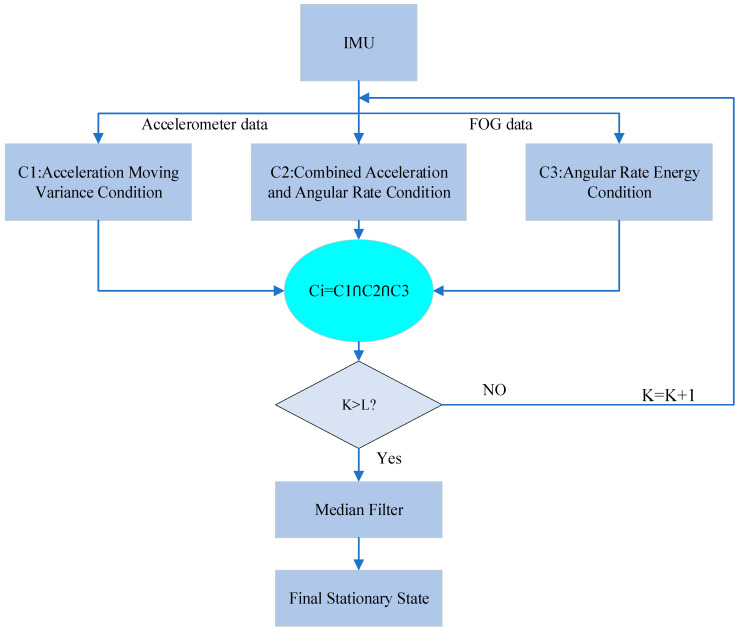
The process of detecting the stationary system-state.

**Figure 3 sensors-24-05911-f003:**
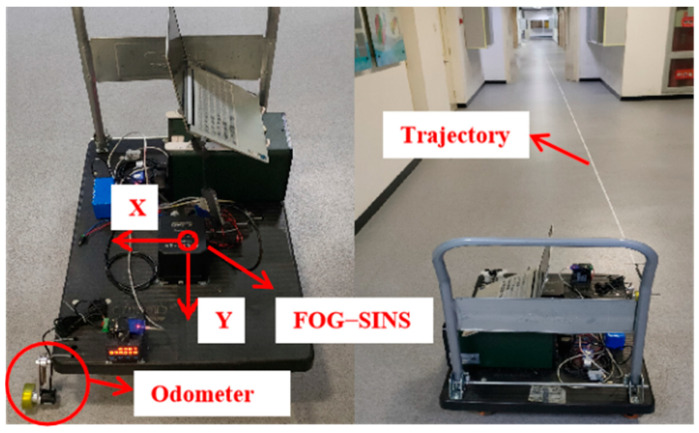
Simulation experiment scene of linear pipe jacking.

**Figure 4 sensors-24-05911-f004:**
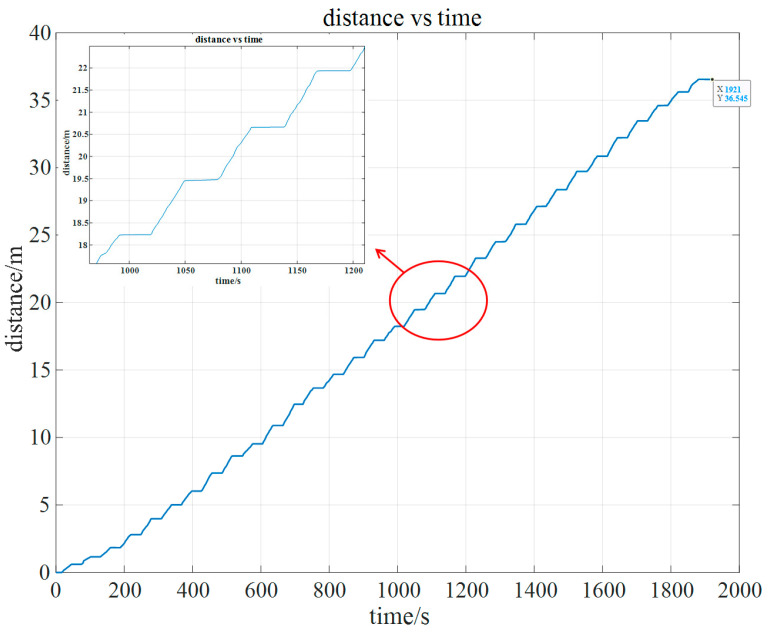
Relationship between distance and time of odometer.

**Figure 5 sensors-24-05911-f005:**
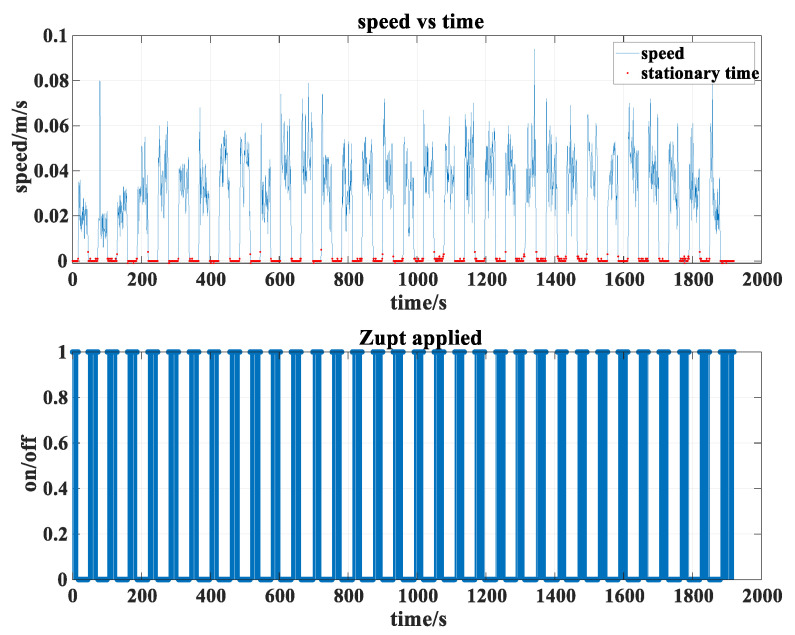
The zero-velocity detection results of the odometer.

**Figure 6 sensors-24-05911-f006:**
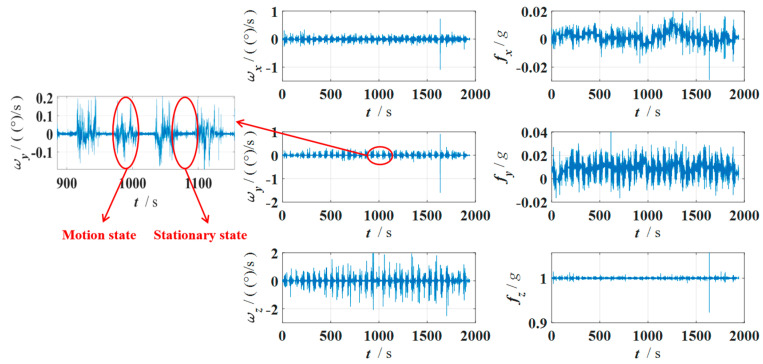
FOG−INS raw data plot of simulated pipe jacking.

**Figure 7 sensors-24-05911-f007:**
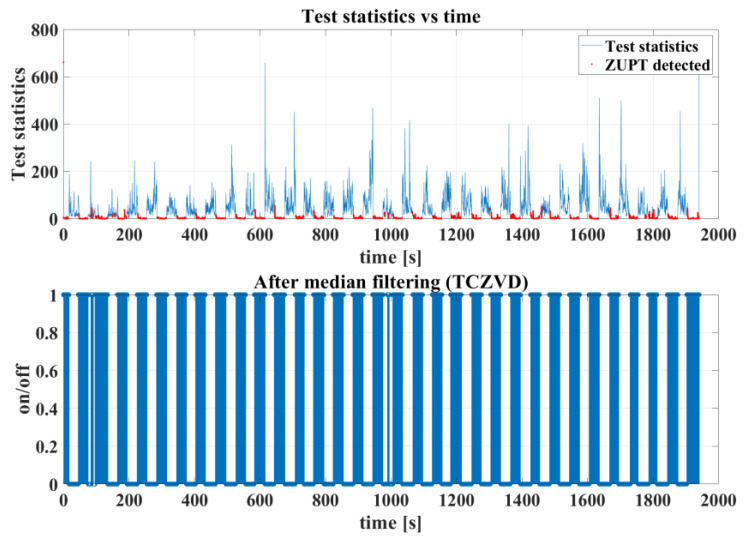
The complete zero-velocity detection results of the TCZVD algorithm.

**Figure 8 sensors-24-05911-f008:**
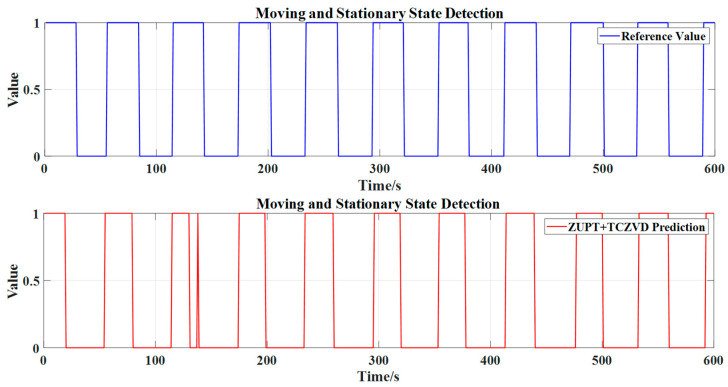
The zero-velocity detection results of TCZVD compared with the reference value.

**Figure 9 sensors-24-05911-f009:**
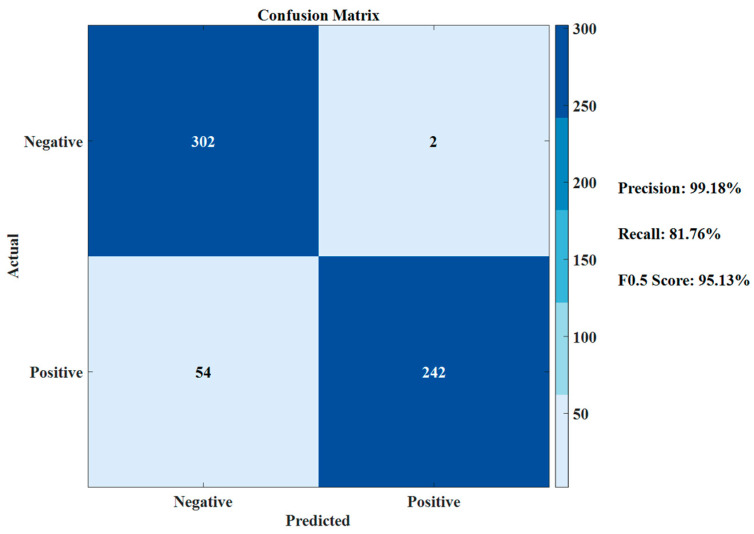
Confusion matrix of the TCZVD algorithm.

**Figure 10 sensors-24-05911-f010:**
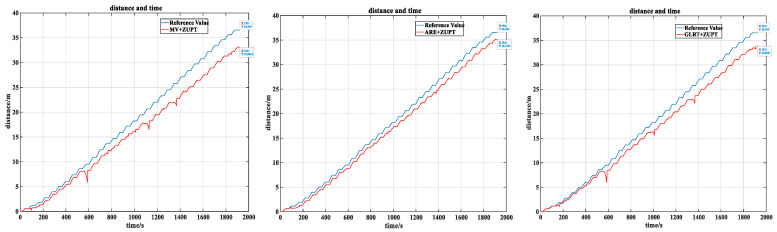
Comparison of position estimation results from commonly used zero-velocity detectors.

**Figure 11 sensors-24-05911-f011:**
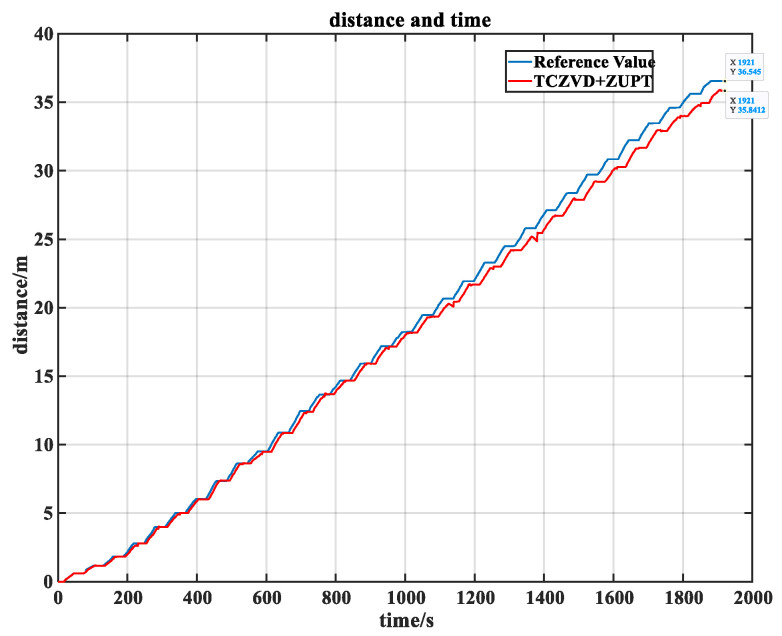
The position estimation results of TCZVD.

**Figure 12 sensors-24-05911-f012:**
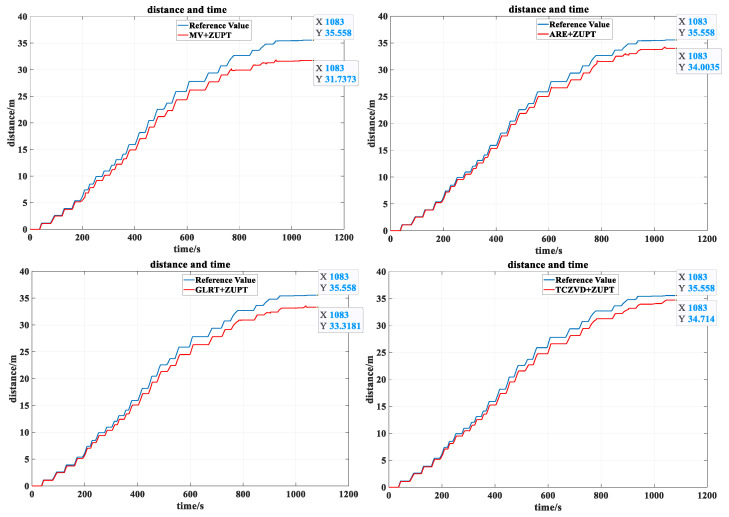
Comparison of position estimation results for each zero-speed detector for different moving and stationary time proportions.

**Figure 13 sensors-24-05911-f013:**
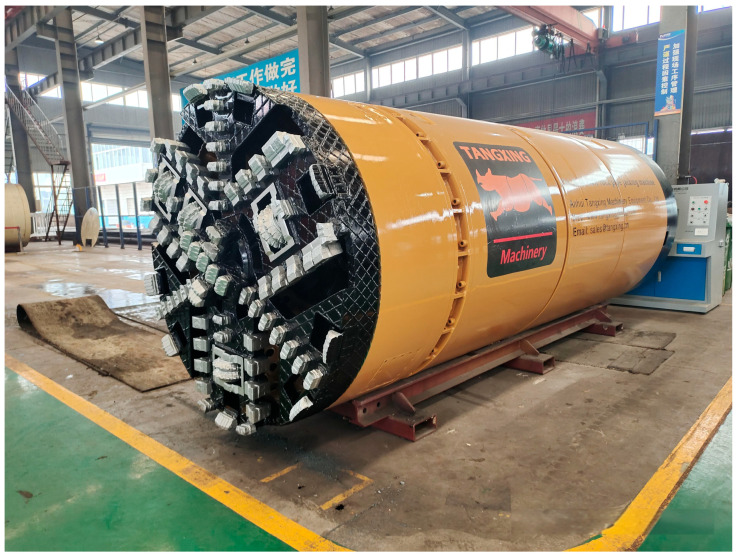
The cutter head of the pipe jacking machine.

**Figure 14 sensors-24-05911-f014:**
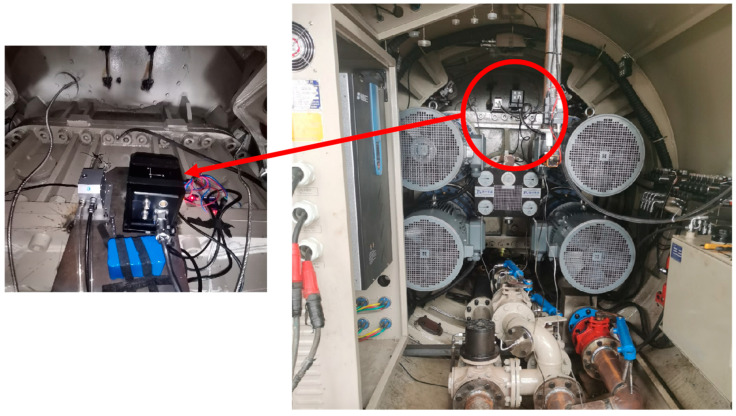
The installation of fiber optic inertial navigation system inside the pipe jacking machine.

**Figure 15 sensors-24-05911-f015:**
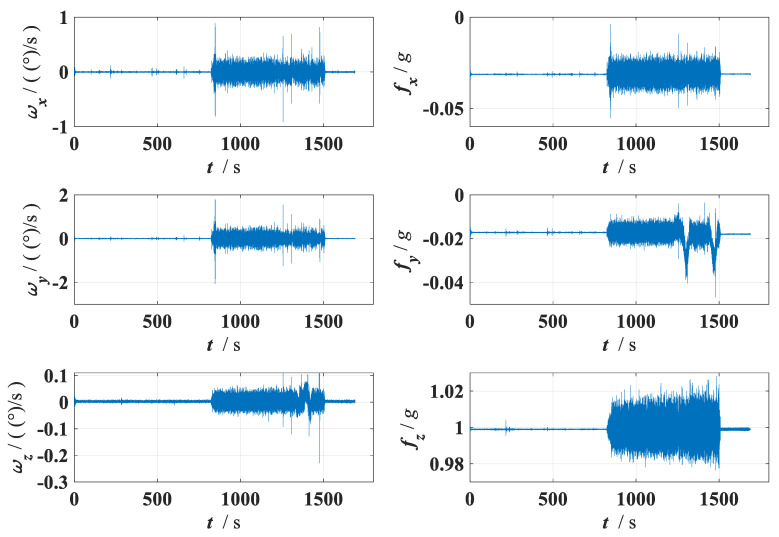
Raw FOG−INS data collected on site from the pipe jacking experiment.

**Figure 16 sensors-24-05911-f016:**
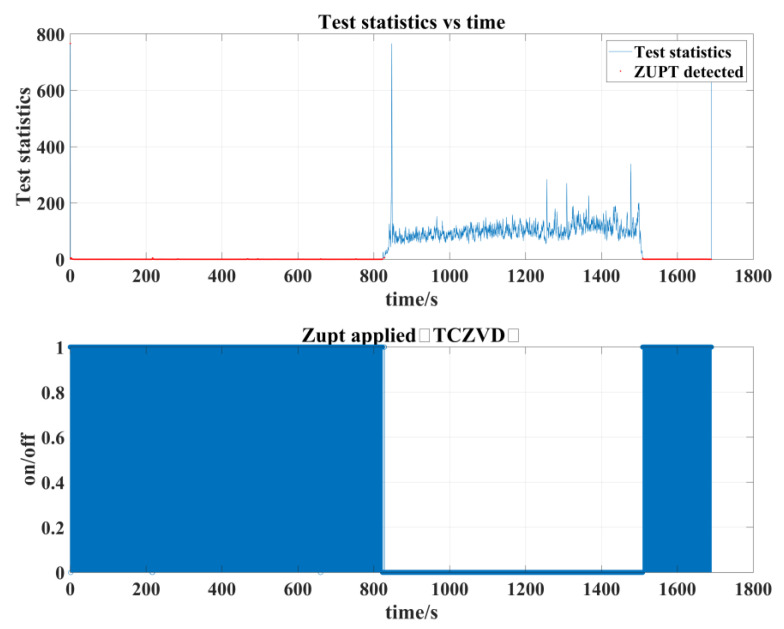
Zero-velocity detection results from the on-site pipe jacking experiment.

**Table 1 sensors-24-05911-t001:** The main parameters of the accelerometer and gyroscope.

Gyroscope		Accelerometer	
Parameters	Value	Parameters	Value
Measurement Range (deg/s)	±1000	Measurement Range (g)	±45
Bias Stability (°/h)	0.5	Bias Stability (mg)	1
Bias Repeatability (°/h)	0.5	Bias Repeatability (mg)	1
Random Walk ($\deg/\sqrt{h}$)	≤0.05	Scale Factor Repeatability (ppm)	≤100
Scale Factor Repeatability (ppm )	≤100		

**Table 2 sensors-24-05911-t002:** Comparison of the position estimation results.

Zero-Velocity Detector	Distance Estimation/m	Position Relative Error/m	Relative Error Rate/%	Accuracy/%
**MV**	33.08	3.47	9.49	90.51
**ARE**	35.17	1.38	3.78	96.22
**GLRT**	33.83	2.72	7.44	92.56
**TCZVD**	35.84	0.71	1.94	98.06

**Table 3 sensors-24-05911-t003:** Comparison of position results for different moving and stationary time proportions of the working conditions.

Zero-Velocity Detector	Distance Estimation/m	Position Relative Error/m	Relative Error Rate/%	Accuracy/%
**MV**	31.74	3.82	10.74	89.26
**ARE**	34.00	1.56	4.39	95.61
**GLRT**	33.32	2.24	6.30	93.70
**TCZVD**	34.71	0.85	2.39	97.61

## Data Availability

Data are contained within the article.

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
