# Peer review of "The Zero-Velocity Correction Method for Pipe Jacking Automatic Guidance System Based on Fiber Optic Gyroscope"

_sensors, 2024, doi:10.3390/s24185911_

Round 1

Reviewer 1 Report

Comments and Suggestions for Authors

1.     1.  It would be helpful to describe the model of inertial algorithms used, as the article only presents a model of an INS error equations, which differs from the models usually used for near-Earth navigation.

2.      It should be helpful to describe the model of INS vertical (altitude) channel, as the model (1)-(3) does not take into account the known instability of the vertical channel.

3.      Models (6)-(9) assume the intersection of conditions for detecting immobility based on accelerometer and gyro readings mainly separately. Why one do not use an integral parameter, such as the scalar product of accelerometer readings and gyroscope readings? It seems that this would be a more effective parameter for detecting INS linear immobility , as it takes into account both accelerometer and gyroscope data..

4.      The authors discuss the effectiveness of zero-velocity correction only. However, mentioned ZUPT aiding do not fundamentally provide information about INS yaw error. Why authors don't take into account the possibility of position aiding using INS relative coordinates? After all, with known lengths of pipeline sections and their number, it is easy to determine the distance traveled by INS. Then it is not difficult to calculate the current INS coordinates with respect to the starting point. Mentioned aiding could be efficient than the ZUPT correction method.

Reviewer 2 Report

Comments and Suggestions for Authors

The current version of the manuscript does not clearly demonstrate significant innovation in the field. The MV, ARE and SHOE zero-velocity detectors have been widely used in the literature. This paper only utilizes the three detectors simultaneously. The experiments also did not demonstrate any advantages over existing methods.

Comments on the Quality of English Language

NONE

Reviewer 3 Report

Comments and Suggestions for Authors

A brief summary

The purpose of this study is to apply Zero Velocity Update (ZUPT) algorithm to improve guidance and positioning accuracy of the Fiber Optic Gyroscope Inertial Navigation System (FOG-INS) for a pipe-jacking machine. This algorithm employs the acceleration and angular rate measurements from the FOG-INS. To detect zero velocity, in this study, Three-Conditional Zero Velocity Detection (TCZVD) algorithm is developed. This algorithm integrates Acceleration-Moving Variance (MV), Angular Rate Energy (ARE), and Stance Hypothesis Optimal Estimation (SHOE) detectors for accurate zero velocity detection. All these algorithms are known, a specific feature is their combination and adaptation to the operating conditions of the pipe-jacking machine.

To evaluate the accuracy of the proposed TCZVD algorithm two sets of experiments are designed. The first set of experiments use a simulation platform to regularly simulate the state changes of jacking and stopping of linear pipe jacking. Experimental results demonstrate that the designed TCZVD algorithm achieves a high zero velocity detection accuracy with positioning errors below 2% of the total distance. The second set of experiments focus on the low-speed environment of practical pipe-jacking operations. Experimental results demonstrate that the proposed algorithm can accurately detect the moments of moving and stationary states of the pipe-jacking machine in practical working conditions.

Specific comments

1.       The results of the experiment with the simulation platform demonstrated positioning errors below 2% of the total distance with an equal ratio of the duration of zero velocity and movement (resting for 30s after every 30s of jacking). It is necessary to predict how the positioning accuracy will change in real operating conditions with a different ratio of zero velocity and movement specified in section 2: "Pipe jacking takes about 2~3h for each segment of pipe (3m), and usually stops for the installation of the next pipe segment. The process of installing a pipe segment typically lasts 10 to 15 minutes".

2.       It is necessary to compare the effectiveness of the various zero velocity detectors (MV, ART and SHOE) for the second set of experiments. According to the graphs shown in Figure 14, it can be said that any of the algorithms under consideration would show high accuracy and that there is no need to use TCZVD.

3.       Text formatting remarks:

a.       In Figure 2 variable L should be described.

b.       δX, H (formulas (2), (5)) and Zk (line 171) must be vectors. The transposition sign may have been omitted.

c.       Figures 6, 14. The labels on the graphs are too small.

d.       Figures 6, 14. The labels on the graphs are not mentioned in the text. Other names are used to indicate acceleration and angular velocity measurements in formulas (6)-(8).

e.       Figures 7, 15. Require clarification of the term "test statistics".

f.        Figure 8. Why do the zero velocity detection results of TCZVD and the reference value have different values on the graph?

g.       Figure 10, Table 2. The figure and the table show the results of the following algorithms: MV, ARE, GLRT, MAG. But only MV, ARE and SHOE are described in the text.

Reviewer 4 Report

Comments and Suggestions for Authors

1.The introduction mentions the "MV+ARE+SHOE" three-conditional zero-speed detection algorithm, which is a major innovation. It is suggested that when elaborating this algorithm, its differences and advantages with the existing methods should be further emphasized to make it more prominent.

2.Some of the long sentences in the manuscript have complex structures. For example, the sentence structure in lines 87-90 is quite complex. It is recommended to break them up appropriately, using shorter or parallel sentences to improve readability and clarity.

3.The introduction uses many technical terms, such as "CNN-ZVD," "MV+ARE+SHOE," etc. It is recommended to provide brief definitions for these terms upon their first occurrence to help readers who may not be familiar with the field.

4.Lines 330-331. Give an explanation as to why only the first 10min of data is taken here for comparative analysis rather than the entire experiment.

5.Some of the specialized terms that appear frequently throughout the manuscript should be given consistent names. For example, moving/stationary state appears several times in the text but is not named consistently.

6.Please carefully check and correct any inconsistencies in tense throughout the manuscript. Ensure that the use of tense in the experimental descriptions, results presentation, and discussion sections is accurate and conforms to the standards of academic writing.

7.The concluding section lacks an outlook for future work, which somewhat diminishes the completeness and forward-looking nature of the paper. It is recommended to add a short discussion of future research directions or potential applications after the conclusion.

Comments on the Quality of English Language

The English expression is good and easy to understand.

Round 2

Reviewer 2 Report

Comments and Suggestions for Authors

According to Figure 2, C1, C2, and C3 are all existing zero-velocity detection methods. The authors' contribution is merely to take the intersection of the three existing detection methods. The innovation of the manuscript is slightly insufficient.

Reviewer 3 Report

Comments and Suggestions for Authors

The purpose of this study is to apply Zero Velocity Update (ZUPT) algorithm to improve guidance and positioning accuracy of the Fiber Optic Gyroscope Inertial Navigation System (FOG-INS) for a pipe-jacking machine. This algorithm employs the acceleration and angular rate measurements from the FOG-INS. To detect zero velocity, in this study, Three-Conditional Zero Velocity Detection (TCZVD) algorithm is developed. This algorithm integrates Acceleration-Moving Variance (MV), Angular Rate Energy (ARE), and Stance Hypothesis Optimal Estimation (SHOE) detectors for accurate zero velocity detection. All these algorithms are known, a specific feature is their combination and adaptation to the operating conditions of the pipe-jacking machine.

To evaluate the accuracy of the proposed TCZVD algorithm two sets of experiments are designed. The first set of experiments use a simulation platform to regularly simulate the state changes of jacking and stopping of linear pipe jacking. Experimental results demonstrate that the designed TCZVD algorithm achieves a high zero velocity detection accuracy with positioning errors below 2% of the total distance. The second set of experiments focus on the low-speed environment of practical pipe-jacking operations. Experimental results demonstrate that the proposed algorithm can accurately detect the moments of moving and stationary states of the pipe-jacking machine in practical working conditions.

In general, all comments are taken into account in the manuscript.

But I cannot agree with the statement that the position error is not related to the regularity of the motion time.

The positioning errors of the INS increase over time and directly depend on the duration of the mode without correction. 

The zero-velocity information is used to correct the velocity error of the INS, but does not completely correct the positioning error accumulated in the non-correction mode.

It is advisable to predict how the positioning accuracy will change in real operating conditions with a different ratio of zero velocity and movement specified in section 2.
